# Mental Health Status of New Police Trainees before and during the COVID-19 Pandemic

**DOI:** 10.3390/healthcare12060645

**Published:** 2024-03-13

**Authors:** Joungsue Kim, Jiyoung Yoon, Inah Kim, Jeehee Min

**Affiliations:** 1General Graduate School, Hanyang University, Seoul 04763, Republic of Korea; sueblue7@naver.com; 2Korea Employment Information Service (KEIS), Eumseong-gun 27740, Republic of Korea; ellie5900@naver.com; 3Department of Occupational and Environmental Medicine, Hanyang University Hospital, College of Medicine, Hanyang University, Seoul 04763, Republic of Korea; inahkim@hanyang.ac.kr

**Keywords:** COVID-19 pandemic, police officers, insomnia, depression, anxiety, mental health

## Abstract

This study aimed to investigate the mental health of new police trainees during the coronavirus disease 2019 (COVID-19) pandemic in Korea. Two groups of police trainees were surveyed considering the distribution of gender, age, and education level: those who joined the school before COVID-19 and those who joined during the outbreak. Mental health indicators, including insomnia, depression, and anxiety, were compared between the two groups. The prevalence of insomnia, depression, and anxiety significantly varied in the group that joined during COVID-19 compared with the group that joined before. Specifically, insomnia showed a significant change in women, with a higher rate of 2.6%. Although the prevalence of depression was initially low, it increased from 0.4% to 1.3% during the pandemic. Anxiety rates also showed notable differences, particularly among women, with a higher rate of 4.7%. The highest differences in prevalence were observed in the low-income group, with a rate of 7.7% for anxiety. The findings highlight the vulnerability of police officers to psychosocial effects during disasters such as pandemics. Disaster preparedness programs or education can be integrated into new police officer training institutions to help manage mental health changes and promote overall well-being.

## 1. Introduction

The duties of the police are to protect the lives, bodies, and property of citizens and to maintain public well-being and order, including the prevention and suppression of crime and investigative work [1]. With the lawful right to use force, police officers’ physical and mental health is crucial for their well-being and community benefit. As first responders, police officers face a work environment laden with traumatic events, violence, armed conflicts, and exposure to serious injuries [2,3]. Within their initial 6 months, officers often encounter an average of three traumatic incidents, with constant exposure to violence, night shifts, and a demanding environment, increasing the risk of stress and psychological issues [4,5]. This high-risk occupational group is susceptible to mental health challenges and physical ailments when compared to the general population [6,7,8].

Research on police officers’ mental health predates coronavirus disease 2019 (COVID-19). Studies on Canadian officers highlighted risks, especially among females, citing factors such as sleep quality and stress [9]. The Buffalo Cardio-Metabolic Occupational Police Stress study revealed a heightened prevalence of insomnia in officers with frequent shift work, consistent with North American studies [10,11]. Acquadro Maran et al.’s 2015 study emphasized the impact of occupational stress and anxiety, advocating for tailored education and support programs [12]. The declaration of COVID-19 as a global public health emergency in January 2020, and its subsequent pandemic status in March, posed unprecedented challenges [13]. The pandemic’s repercussions included adverse social and economic effects, leading to a surge in psychological issues such as panic disorders, anxiety, and depression [14,15]. Police officers, frontline responders managing the pandemic, faced heightened risks of psychosocial factors, anxiety, and depression [16,17].

The police academy is vital for a police career, providing skills, knowledge, and preparation for challenges [18,19]. The COVID-19 pandemic exposed deficiencies in digital readiness, equipment, and infrastructure for police training, revealing numerous challenges ahead [20]. The Central Police Academy (CPA) in Korea adapted its education system and rules due to COVID-19, combining face-to-face and remote education, implementing mask-wearing, and restricting activities based on social distancing guidelines. New police officer trainees, especially those trained during the pandemic, often felt disconnected from exclusively remote classes [21,22]. Additionally, the pandemic hindered on-the-job training, impacting the adequacy of training for these new recruits compared to previous practices. As such, the COVID-19 pandemic is profoundly reshaping the approach and practices in the area of law enforcement training [23]. 

This study aimed to assess the mental health of new police officers before their recruitment and to monitor changes resulting from occupational factors after they joined the service. The cohort study was conducted as part of a long-term cohort study on the mental health of new police officers. Meanwhile, the cohort sub-study was conducted to investigate changes in trainees’ mental health due to external influences to explore the mental health status of new police officer trainees in Korea during the COVID-19 pandemic. Quantifying the impact of COVID-19 on mental health is imperative for informed crisis response. Moreover, understanding COVID-related stress is crucial for effectively managing high-risk police officers in need of mental health care during a public health crisis [24]. Therefore, the results from this study are important because they may reflect the ability of new police officers to respond to crisis situations once they become active police officers. Furthermore, we aim to contribute to policies that manage and prevent mental disorders in high-risk groups and lay the foundation for the discovery of associations through follow-up surveys.

## 2. Materials and Methods

### 2.1. The Study Population

The new police officer survey includes new police officers who are in the process of receiving training at the CPA before starting field jobs after passing the selection examination. These are new police officers selected from across the country who have similar knowledge, education, and physical requirements. In this study, the first respondents were trainees who entered the school before the COVID-19 pandemic (August 2019 baseline survey), and the second respondents comprised trainees who joined during the COVID-19 (October 2020 baseline survey). The total population was 2898 people in 2019 and 2726 in 2020. Police officers who did not consent to the study, those who insincerely answered the questionnaire, and those with missing data on sex and age were excluded. Finally, 2488 (85.9%) in 2019 and 2465 (90.4%) in 2020 were included in this study. The high response rate is indicative of the main tendencies of the students of the CPA.

### 2.2. Data Collection

Participants were recruited through a self-administered survey of all CPA students enrolled in theoretical training at the time of the survey (two times: August 2019, October 2020). They entered the CPA, the representative recruitment gateway for the Korean police, and received approximately 8 months of training, which included theoretical and field training, before being hired. Among all respondents, those who did not agree to participate in the study or did not truthfully answer the mental health questionnaire were excluded. The survey included socioeconomic status, such as sex, age, marital status, education level, and monthly household income. We asked regarding any past medical history of mental illness, although no one reported a history of mental illness. In addition, mental health information obtained included depression, anxiety, insomnia, stress index, and resilience.

### 2.3. Outcome Measurement

In this study, the level of mental health was classified into insomnia, depression, anxiety, and perceived stress, measured using a self-reported method. The tools used in the study were as follows. First, the Korean version of the insomnia severity index (ISI-K) was used as an insomnia tool to determine the presence of insomnia [25,26]. The test tool consists of seven items measuring the severity of insomnia. A five-point scale consisting of 0–4 points is used, and a score of 15 or more is considered a moderate level of insomnia. Second, to assess the presence of depression in the study participants, the researchers utilized the Patient Health Questionnaire-9 (PHQ-9) [27]. The Korean version of PHQ-9 consists of nine items that measure depression in adults. A four-point scale consisting of 0–3 points is used, and a score of 10 or higher is judged to be moderate or more depressed [28]. Finally, for anxiety, Generalized Anxiety Disorder-7 (GAD-7) was used [29]. The Korean version of GAD-7 consists of seven items measuring anxiety in adolescents and adults. The score uses a four-point scale consisting of 0–3 points, and a score of 5 or higher means a mild anxiety level [30]. Mental health levels were recorded as binary variables based on the result interpretation criteria of each tool. Afterward, the difference in mental health before and after COVID-19 was confirmed based on the prevalence difference by the group. All analyses were stratified by sex, and subgroup analysis was performed according to age, education level, and household income.

### 2.4. Participant Demographics 

Participants were grouped according to age (20–29 and over 30 years old), sex (men or women), marital status (single, married), educational group (under college graduate, college graduate, or higher), and monthly household income (less than 4000 and more than 4000 USD). Regarding marital status, individuals who were not currently married were classified as single. The level of education was determined by graduation from a college; all participants enrolled in college or on leave of absence were classified as having an education lower than a graduate degree. Monthly household income was categorized according to the average monthly household income in the Republic of Korea. 

### 2.5. Statistical Analysis

The prevalence differences in mental health according to sex were also stratified and analyzed. First, a frequency analysis was performed on the demographic background. The categorical variable reported the count and percentage, and the continuous variable reported the mean and standard deviation. Demographic characteristics were compared between the study population from 2019 and those from 2020 using Chi-squared tests. Also, the prevalence of mental illness (Insomnia, Depression, Anxiety) and 95% confidence interval (CI) were calculated, and the prevalence difference and 95%CI for the difference between the two periods were calculated using a binomial distribution. Subgroup analyses were performed by adjusting for age, education, and household income. In the analyses, considering the distribution of participants, age was divided into 20s and over 30s, education level was divided into less than a college degree or less and college degree or higher, and household income was divided into approximately less than 4000 USD or less and about 4000 USD or higher. The calculation of the prevalence and the prevalence difference was analyzed using the Generalized Linear Model: Extensions to the Binomial Family (‘binreg’ in STATA). For the comparability of the two study groups, we performed propensity score matching (PSM) using age, sex, education, and monthly household income. 

All statistical analyses were reported as two-tailed tests, and statistical significance was set at *p* < 0.05. Analysis was performed using STATA version 17.0 (STATA Institute Inc., Cary, NC, USA).

## 3. Results

The demographic characteristics of the study participants in 2019 and 2020 are shown in Table 1. The total number of participants was 2488 in 2019 (1806 men [72.6%] and 682 women [27.4%]) and 2465 in 2020 (1793 men [72.7%] and 672 women [27.3%]), with mean ages of 28.0 (±0.1) and 25.6 (±0.1), respectively. Distributions in terms of sex, age groups, marriage, education level, and household income were similar between the two survey periods. When analyzed using the Chi-squared test, the differences in the distributions of the age and income variables among the group variables in two survey periods were statistically significant, while the differences in the distributions of the other variables, such as sex, marriage, education level, and household income, were not statistically significant. Since the groups were not identical, we performed further analysis using the PSM method; however, there was no variance from the original data analysis outcomes; therefore, only the raw data analysis results are presented (Appendix A).

The prevalence of insomnia in the study participants is shown in Table 2. The prevalence of insomnia significantly increased by 0.9% (95%CI, 0.0–1.8), from 2.0% in 2019 to 2.9% in 2020 (Table 2). Education level significantly increased by 1.8% (0.4–3.2), from 1.8% in 2019 to 3.6% in 2020 for those with a college degree or higher. Regarding monthly household income, the group with less than approximately 4000 USD had a 1.5% increase (0.2–2.9), from 2.3% in 2019 to 3.8% in 2020, which was statistically significant. The subgroup analysis showed that monthly household income significantly increased among women by 2.6% (0.4–4.7), but no statistically significant difference was observed among men. In addition, education level in women statistically significantly increased by 4.2% (1.2–7.2), from 2.8% in 2019 to 6.9% in 2020 for those with a college degree or higher.

Table 3 shows that the prevalence of depression significantly increased by 0.9% (0.4–1.4), from 0.4% in 2019 to 1.3% in 2020. Age (0.8% in those in their 20s), education level (1.7% in those with a college degree or higher), and monthly household income (1.5% in those who earn less than 4000 USD) also significantly increased. In the subgroup analysis, the prevalence of depression significantly increased in women (1.5%, 0.2–2.8) than in men (0.7%, 0.2–1.1). For men, age group (0.7% in those in their 20s), education level (1.3% in those with a college degree or higher), monthly household income (1.3% in those who earn less than 4000 USD) statistically significantly increased, although there was a difference in the degree in each group. However, in the case of women, there was only a 2.5% (0.3–4.7) increase in the group with a college degree or higher.

Table 4 shows that the prevalence of anxiety significantly increased by 2.0% (1.1–2.8), from 1.2% in 2019 to 3.2% in 2020. There was a significant difference in age (2.0% in those aged 20–29 years), education level (1.3% in those with less than a college degree vs. 2.9% in those with a college degree or higher), and monthly household income (3.5% in those earning less than 4000 USD). In the subgroup analysis, the prevalence of anxiety significantly increased in women (4.7%, 2.3–6.9) than in men (1.0%, 0.2–1.7). The increase in anxiety in women was greater, but in men, it significantly increased in all groups except in those aged over 30 years and those earning 4000 USD or higher.

## 4. Discussion

In this study, we investigated changes in the mental health of new police trainees, considering alterations in the educational environment of the Central Police Academy and broader societal shifts caused by COVID-19. The findings indicated an increase in insomnia, depression, and anxiety symptoms among police trainees, attributed to external factors such as a pandemic. Furthermore, the study observed significantly higher levels of insomnia, depression, and anxiety in women, whereas insomnia did not exhibit statistical significance in men. These outcomes align with previous mental health research highlighting women’s vulnerability [31,32] and the documented impact of the COVID-19 pandemic on mental health [33].

The mental health of the police trainees was not an actively discussed topic. Nevertheless, there has been some discussion about the level of mental health as a leading indicator of how people will cope with high job stress in the future. A study of the mental health of police apprentices in Sweden found that they were mentally much healthier than the general population. In terms of personality, Harm Avoidance and Self Directedness were found to be associated with all aspects of psychopathology, as well as good mental health and mature personality traits, with women reporting more of these traits [19]. In the other Swedish study, police recruits self-reported being happier, less anxious, less aggressive, and less impulsive than civilians; less sensitive to reinforcement, whether in terms of punishment or reward; and more socially desirable [34]. A longitudinal survey of 103 Swedish police trainees to study personality changes and mental health responses during the first 2 weeks of enlistment confirmed several personality changes [35].

The prevalence of insomnia among new police trainees was 2.9%, lower than the 11.2% prevalence of moderate to severe insomnia during COVID-19 in a cross-sectional study of first responders in China [36] using the same tool, but with a prevalence of 0.9% compared to before the pandemic, which is a statistically significant increase. In particular, insomnia worsened more prominently in women at 2.6%, which is the same as the result of gender differences in mental disorders in Canadian police officers [9]. In addition, after a new police officer graduates from the Central Police School and is deployed to the site, the quality of sleep is likely to worsen due to a wide range of occupational factors, such as long working hours, shift work, and stress [10]. Untreated sleep disorders in police officers are indirectly related to PTSD, depression, and anxiety symptoms [9] and can adversely affect mental health and safety and pose a risk to the public [11]; therefore, they should be detected and diagnosed early with regular assessments to prevent the risk.

The prevalence rate of Moderate Major Depressive Disorder (MDD) was higher in women (0.9%, 2.4%) than in men (0.2%, 0.8%) in 2019 and 2020, respectively. These results are lower than the prevalence of MDD in the general population aged over 19 years in Korea during the COVID-19 pandemic (men: 4.4%, women: 6.2%) [37] attributed to age distribution (average age; 2019: 28.0 ± 0.1, 2020: 25.6 ± 0.1) and a selectively healthier group. However, this is consistent with existing literature that female police officers are more prone to MDD and PTSD diagnoses than men [38]. In contrast to a Korean study where the prevalence of MDD was higher in women, but the change before and after COVID-19 was higher in men [37], our study revealed a statistically significant increase in depression prevalence in both men (0.7% 0.2–1.1) and women (1.5%, 0.2–2.8), with a greater change observed in women. This supports previous research findings that the mental health risk of police officers during the pandemic was higher in women [16]. Furthermore, low-income status, a recognized risk factor for MDD, intensified prevalence, excluding females, consistent with prior findings [39]. However, contrary to past research, there was no protective effect of education against depression. It is speculated that coping strategies, such as personal coping skills, exerted a greater impact on depression than the protective effect of education among subjects of similar age and occupation. Moreover, while the study population was categorized based on graduation from a college, only 5.4% of the participants had less than a high school diploma, possibly because most subjects pursued higher education at the college or university level.

Although the prevalence of Generalized Anxiety Disorder is estimated to be 1.6% to 5.0% in the general population [40], the prevalence of mild anxiety or higher in this study was 0.7% (2019, male) to 7.1% (2020, female), and the highest difference in prevalence was shown in females (4.7%) during the pandemic period. In contrast to the Chinese study reporting an average anxiety score of 3.6 ± 4.2 and a prevalence of 8.8% for moderate to severe anxiety [41], participants in our study had low average anxiety scores (0.4 ± 1.1 in 2019, 0.6 ± 1.7 in 2020), and the prevalence of anxiety was relatively low. These results align with an Australian study indicating that women, youth, and low income contribute to increased anxiety [42]. They are also consistent with a study reporting higher psychological distress and organizational stress levels among female police officers in all roles compared to their male counterparts [12]. Additionally, anxiety, like depression, showed an association with greater variation in the highly educated group. Therefore, caution is advised when interpreting the socioeconomic status of the participants in this study. Anxiety symptoms have the potential to lead to various psychological problems, hinder the adoption of effective coping strategies [12], and warrant careful observation, as they are likely to co-occur with depression. 

Even in a sample of police officers who were considered to have better resilience to the psychiatric sequelae of trauma than other first responders [43], mental health indicators, such as PTSD, depression, and anxiety symptoms, remained high for a long time after a disaster [44]. Additionally, nearly half of the police officers with probable PTSD reported having comorbid mental health problems, such as symptoms of anxiety and depression, which may explain chronic impairment in mental health and social functioning [43]. To gain a broader view of the increase in mental health problems during the pandemic, several methodological, response-related, pandemic-related and health-policy-relevant factors must be considered to identify detrimental factors for mental health [45], which are evidence-based to systematically respond to changes in mental health in other future disaster situations. This study can be used as a basis for predicting and preventing mental health deterioration in disaster situations and can provide important information for the mental health management process from police trainees to officers.

This study has some limitations. First, this study assessed the absolute difference in prevalence by comparing two cross-sectional data sources. Therefore, a longitudinal study is needed to confirm the causal association and temporal relationship in the future. Second, this study used a self-administered questionnaire and was not diagnosed by a doctor. This has the limitation of under-reporting clinical symptoms in the self-report questionnaire. However, a health questionnaire using a verified health scale was used, and potential measurement errors of outcome variables could be reduced due to a relatively large sample. Third, exposure to COVID-19 stressors received in society before entering the Central Police Academy cannot be distinguished. This reflects the lack of regular mental health screenings from new police officer trainees to police officers. Therefore, the implementation of more regular mental health screenings should provide evidence to collectively identify those in need of psychological support. Fourth, in this study, we could not consider variable confounding factors. Due to the nature of mental health, an individual’s ability to cope is not only related to their socioeconomic resources, but also to their innate features, such as personality. However, in this study, we could not assess the traits of an individual’s personality and individual resilience. Finally, since the participants of this study were new police officer trainees who were selected after undergoing a healthy physical and mental health evaluation, there may be a biased health worker effect compared to the general population, and there is a selection bias pertaining to a relatively young average age. Despite these limitations, this study has sufficient meaning as the first study to evaluate the basic survey of mental health and changes according to disaster situations targeting new police trainees in Korea. In addition, the response rate of the survey is very high; therefore, it is representative data by which to understand the annual baseline mental health status of new police officers. Based on this study, among other things, the deterioration in the mental health status of new police officers who joined after COVID-19 was revealed. This study’s results will help establish health policies related to police mental health.

## 5. Conclusions

In conclusion, long-term deterioration of the mental health of police officers and stressful situations has an important public health effect on the stability of the entire community, and society has a high influence on improving the health of police officers.

Education regarding stress management and psychosocial problems should be provided to first responders and leaders of the medical system, as well as medical professionals [46], and has an important role in the training process for new police officers. The mental health of new officers is also important as a leading indicator of how they may adapt to high-stress environments in the future. In particular, police officers in disaster situations, such as pandemics, may be more vulnerable to psychosocial effects due to increased stress while working; therefore, mental health problems are emphasized, and mental health changes can be managed through disaster preparedness programs or disaster preparedness education at new police officer training institutions, suggesting the need for educational and policy measures. In addition, as this study is the first investigation in Korea into the psychological impact of COVID-19 on new police officers, it can be used as a criterion for examining factors and the degree of influence on changes in mental health status.

## Figures and Tables

**Table 1 healthcare-12-00645-t001:** Socio-demographic characteristics in the new police officer trainee survey 2019 and 2020 samples stratified by sex.

	2019		2020		2019				2020			
	Total (N = 2488)		Total (N = 2465)		Men (N = 1806)		Women (N = 682)		Men (N = 1793)		Women (N = 672)	
	N (%)	CI	N (%)	CI	N (%)	CI	N (%)	CI	N (%)	CI	N (%)	CI
Average Age (M, SD)	28.0 (±0.1)	27.8–28.1	25.6 (±0.1)	25.4–25.7	28.2 (±0.1)	28.1–28.4	27.2 (±0.1)	27.0–27.5	25.8 (±0.1)	25.7–26.0	24.9 (±0.1)	24.6–25.1
Age												
20–29	1823 (73.3)	71.5–75.0	2190 (88.8)	87.5–90.0	1279 (70.8)	68.7–72.9	544 (79.8)	76.6–82.3	1581 (88.2)	86.6–89.6	609 (90.6)	88.2–92.6
≥30	665 (26.7)	25.0–28.5	275 (11.2)	10.0–12.5	527 (29.2)	27.1–31.3	138 (20.2)	17.4–23.4	212 (11.8)	10.4–13.4	63 (9.4)	7.4–11.8
Marriage												
Not Married	2434 (97.8)	97.2–98.3	2420 (98.2)	97.6–98.6	1764 (97.7)	96.9–98.3	670 (98.2)	96.9–99.0	1768 (98.6)	97.9–99.1	652 (97.0)	95.4–98.1
Married	52 (2.1)	1.6–2.7	43 (1.7)	1.3–2.3	42 (2.3)	1.7–3.1	10 (1.5)	0.8–2.7	23 (1.3)	0.9–1.9	20 (3.0)	1.9–4.6
missing value	2 (0.1)	0.0–0.3	2 (0.1)	0.0–0.3	0 (0.0)	–	2 (0.3)	0.1–1.2	2 (0.1)	0.0–0.4	0 (0.0)	–
Education												
<College degree	1317 (52.9)	51.0–54.9	1435 (58.2)	56.3–60.1	1037 (57.4)	55.1–59.7	280 (41.1)	37.4–44.8	1138 (63.5)	61.2–65.7	297 (44.2)	40.5–48.0
≥College degree	1168 (47.0)	45.0–49.0	1027 (41.7)	39.7–43.6	768 (42.5)	40.3–44.8	400 (58.6)	54.9–62.3	652 (36.3)	34.2–38.6	375 (55.8)	52.0–59.5
missing value	3 (0.1)	0.0–0.4	3 (0.1)	0.0–0.4	1 (0.1)	0.0–0.4	2 (0.3)	0.1–1.2	3 (0.2)	0.1–0.5	0 (0.0)	–
Monthly household income												
<about 4000 USD	1314 (52.8)	50.8–54.8	1170 (47.5)	45.5–49.4	1014 (53.2)	53.8–58.4	300 (44.0)	40.3–47.7	871 (48.6)	46.3–50.1	299 (44.5)	40.8–48.3
≥about 4000 USD	1141 (45.7)	43.9–47.8	1244 (50.5)	48.5–52.4	772 (42.8)	40.5–45.0	369 (54.1)	50.3–57.8	889 (49.6)	47.3–51.9	355 (52.8)	49.0–56.6
missing value	33 (1.3)	0.9–1.9	51 (2.1)	1.6–2.7	20 (1.1)	0.7–1.7	13 (1.9)	1.1–3.3	33 (1.8)	1.3–2.6	18 (2.7)	1.7–4.2

**Table 2 healthcare-12-00645-t002:** Prevalence of insomnia in the 2019 sample (N = 2488) and in the 2020 sample (N = 2465) in Korea.

Insomnia		Total			Men			Women		
		2019	2020	Difference	2019	2020	Difference	2019	2020	Difference
		(N = 2488)	(N = 2465)	(95%CI)	(N = 1806)	(N = 1793)	(95%CI)	(N = 682)	(N = 672)	(95%CI)
Total		2.0	2.9	0.9 *	1.7	2.0	0.3	2.9	5.5	2.6 *
		(1.5–2.6)	(2.3–3.7)	(0.0–1.8)	(1.1–2.4)	(1.4–2.7)	(−0.6–1.2)	(1.8–4.5)	(3.9–7.5)	(0.4–4.7)
Age group	20–29	2.2	2.9	0.7	1.8	1.9	0.1	3.1	5.4	2.3
		(1.6–3.0)	(2.2–3.7)	(−0.3–1.7)	(1.1–2.7)	(1.3–2.7)	(−0.9–1.1)	(1.8–5.0)	(3.8–7.5)	(−0.0–4.6)
	≥30	1.5	3.3	1.8	1.3	2.4	1.0	2.2	6.3	4.2
		(0.7–2.7)	(1.5–6.1)	(−0.5–0.4)	(0.5–2.7)	(0.8–5.4)	(−1.2–3.2)	(0.5–6.2)	(1.8–15.5)	(−2.3–10.7)
Education group	<College degree	2.2	2.4	0.2	1.9	2.1	0.2	3.2	3.7	0.5
		(1.5–3.1)	(1.7–3.4)	(−0.9–1.4)	(1.2–3.0)	(1.4–3.1)	(−1.0–1.4)	(1.5–6.0)	(1.9–6.5)	(−2.5–3.5)
	≥College degree	1.8	3.6	1.8 *	1.3	1.7	0.4	2.8	6.9	4.2 *
		(1.1–2.7)	(2.5–4.9)	(0.4–3.2)	(0.6–2.4)	(0.8–3.0)	(−0.9–1.7)	(1.4–4.9)	(4.6–10.0)	(1.2–7.2)
Monthly household income	<4000 USD	2.3	3.8	1.5 *	1.9	2.9	1.0	3.7	6.7	3.0
		(1.5–3.2)	(2.8–5.1)	(0.2–2.9)	(1.1–2.9)	(1.9–4.2)	(−0.4–2.4)	(1.8–6.5)	(4.1–10.1)	(−0.1–6.6)
	≥4000 USD	1.8	2.2	0.4	1.4	1.1	–0.3	2.4	4.8	2.3
		(1.1–2.7)	(1.4–3.1)	(−0.7–1.5)	(0.7–2.5)	(0.5–2.1)	(−1.0–0.8)	(1.1–4.6)	(2.8–7.6)	(−0.4–5.1)

Values are presented as % (95%CI). CI, confidence interval (95%CI). * *p* < 0.05.

**Table 3 healthcare-12-00645-t003:** Prevalence of depression in the 2019 sample (N = 2488) and in the 2020 sample (N = 2465) in Korea.

Depression		Total			Men			Women		
		2019	2020	Difference	2019	2020	Difference	2019	2020	Difference
		(N = 2488)	(N = 2465)	(95%CI)	(N = 1806)	(N = 1793)	(95%CI)	(N = 682)	(N = 672)	(95%CI)
Total		0.4	1.3	0.9 *	0.2	0.8	0.7 *	0.9	2.4	1.5 *
		(0.2–0.7)	(0.9–1.8)	(0.4–1.4)	(0.0–0.5)	(0.5–1.4)	(0.2–1.1)	(0.3–1.9)	(1.4–3.8)	(0.2–2.8)
Age group	20–29	0.4	1.3	0.8 *	0.2	0.9	0.7 *	0.9	2.3	1.4
		(0.2–0.9)	(0.9–1.8)	(0.3–1.4)	(0.0–0.7)	(0.5–1.5)	(0.1–1.2)	(0.3–2.1)	(1.3–3.8)	(−0.0–2.8)
	≥30	0.2	1.1	0.9	0.0	0.5	0.0^a^	0.7	3.2	2.4
		(0.0–0.8)	(0.2–3.2)	(−0.0–2.2)	(0.0–0.7)	(0.0–2.6)	0.0–0.0	(0.0–4.0)	(0.4–11.0)	(−0.0–7.0)
Education group	<College degree	0.2	0.6	0.3	0.2	0.5	0.3	0.4	0.7	0.3
		(0.0–0.7)	(0.2–1.1)	(−0.0–0.8)	(0.0–0.7)	(0.2–1.1)	(−0.0–0.8)	(0.0–2.0)	(0.0–2.4)	(−0.8–1.5)
	≥College degree	0.5	2.2	1.7 *	0.1	1.4	1.3 *	1.3	3.7	2.5 *
		(0.2–1.1)	(1.4–3.3)	(0.7–2.7)	(0.0–0.7)	(0.6–2.6)	(0.3–2.2)	(0.4–2.9)	(2.1–6.2)	(0.3–4.7)
Monthly household income	<4000 USD	0.2	1.7	1.5 *	0.0	1.4	1.3 *	0.7	2.7	2.0
		(0.0–0.7)	(1.0–2.6)	(0.7–2.3)	(0.0–0.5)	(0.7–2.4)	(0.5–2.1)	(0.1–2.4)	(1.2–5.2)	(−0.0–4.1)
	≥4000 USD	0.5	0.9	0.4	0.3	0.3	0.0	1.1	2.3	1.2
		(0.2–1.1)	(0.4–1.6)	(−0.3–1.0)	(0.0–0.9)	(0.1–1.0)	(−0.4–0.6)	(0.3–2.8)	(1.0–4.4)	(−0.7–3.0)

Values are presented as % (95%CI). CI, confidence interval (95%CI). * *p* < 0.05.

**Table 4 healthcare-12-00645-t004:** Prevalence of anxiety in the 2019 sample (N = 2488) and in the 2020 sample (N = 2465) in Korea.

Anxiety		Total			Men			Women		
		2019	2020	Difference	2019	2020	Difference	2019	2020	Difference
		(N = 2488)	(N = 2465)	(95%CI)	(N = 1806)	(N = 1793)	(95%CI)	(N = 682)	(N = 672)	(95%CI)
Total		1.2	3.2	2.0 *	0.7	1.7	1.0 *	2.5	7.1	4.7 *
		(0.8–1.7)	(2.5–3.9)	(1.1–2.8)	(0.4–1.0)	(1.1–2.4)	(0.2–1.7)	(1.5–4.0)	(5.3–9.4)	(2.3–6.9)
Age group	20–29	1.2	3.2	2.0 *	0.8	1.7	0.9 *	2.0.	6.9	4.9 *
		(0.7–1.8)	(2.5–4.0)	(1.1–2.9)	(0.4–1.4)	(1.1–2.5)	(0.1–1.7)	(1.0–3.6)	(5.0–9.2)	(2.5–7.2)
	≥30	1.4	3.3	1.9	0.6	1.4	0.8	4.3	9.5	5.2
		(0.1–2.6)	(1.5–6.1)	(−0.4–0.4)	(0.1–1.7)	(0.3–4.1)	(−0.9–2.6)	(1.6–9.2)	(3.6–19.6)	(−2.8–13.2)
Education group	<College degree	1.0	2.3	1.3 *	0.8	1.7	0.9	1.8	4.7	2.9 *
		(1.0–1.7)	(1.6–3.2)	(0.4–2.2)	(0.3–1.5)	(1.0–2.6)	(−0.0–1.8)	(0.6–4.1)	(2.6–7.8)	(0.1–5.8)
	≥College degree	1.5	4.4	2.9 *	0.7	1.7	1.0	3.0	9.1	6.1 *
		(0.9–2.3)	(3.2–5.8)	(1.5–4.4)	(0.2–1.5)	(0.8–3.0)	(−0.1–2.2)	(1.6–5.2)	(6.4–12.4)	(2.7–9.4)
Monthly household income	<4000 USD	1.2	4.7	3.5 *	0.8	2.8	2.0 *	2.7	10.4	7.7 *
		(0.7–2.0)	(3.6–6.1)	(2.1–4.8)	(0.3–1.5)	(1.8–4.1)	(0.8–3.2)	(1.1–5.2)	(7.2–14.4)	(3.8–11.6)
	≥4000 USD	1.2	1.8	0.5	0.6	0.7	0.0	2.4	4.5	2.1
		(0.7–2.1)	(1.1–2.7)	(−0.4–1.5)	(0.2–1.5)	(0.2–1.5)	(−0.8–0.8)	(1.1–4.6)	(2.6–7.2)	(−0.6–4.7)

Values are presented as % (95%CI). CI, confidence interval (95%CI). * *p* < 0.05.

## Data Availability

The data are available upon request. The dataset contains personal, identifiable, and sensitive information. Therefore, we were not allowed to make them publicly available, according to the Personal Information Protection Commission in Republic of Korea.

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
