# Peer review of "Mental Health Status of New Police Trainees before and during the COVID-19 Pandemic"

_healthcare, 2024, doi:10.3390/healthcare12060645_

Round 1

Reviewer 1 Report

Comments and Suggestions for Authors

Thank you for the opportunity to revise the article. The results are very important from a practical perspective. However, there are some issues that need to be addressed:

- what about other genders? What are the regulations in Korea regarding this issue?

- some typos (line 100- double bracket)

- it's not clear to me how the income was counted. If I understand correctly, the candidates planned to become police officers. In their 20s, without previous experiences, would they have any income (except the social status of their parents)?

- I've got the impression that some factors that could influence the analysis of anxiety and depression are missed. Those are, e.g., the personality traits of trainees. In some countries, the trainees are examined on this- maybe the Authors have access to this data type?

- little is know about previous metal health problems history of the participants

Reviewer 2 Report

Comments and Suggestions for Authors

 1.                The introduction does not clearly justify the need to carry out this study, why the study is really being carried out. I get the impression that they give all the police officers who enter the academy the questionnaires and once they have the data, they have decided that we are going to compare the group before and the group during the COVID-19 pandemic. If they wanted to know how the pandemic influenced the new police officers, I think it would have been more useful to have studied the same group of police officers at different times and to have seen the changes in terms of mental health. Therefore, I think it is necessary to justify the need for the study much better and to base it on the existing literature.

2.                In terms of methodology, I think it is important to include important aspects in order to understand the study:

a.                There is no section on the procedure followed in the research. The way the study is presented, it is very difficult, if not impossible, to repeat it in another context.

b.               It is not specified how the police officers who made up the study were selected. They were passed on to all police officers. A sample was taken. I do not know.

c.                It is also not detailed when the questionnaires were given to the police officers. If it was when they entered the academy, if it was when they had been there for a while, if it was when the pandemic was going on, etc.

d.               They do not indicate the criteria they used to group the sample into the age brackets indicated in the study.

3.                In the results section:

a.                A table 2 appears which is not referenced in the text.

b.               I think that some analysis should have been carried out to see if the groups were as homogeneous as possible in order to make comparisons between them.

c.                I also think that in order to be able to make reliable comparisons, a pre-test and a post-test should have been carried out in each group.

d.               Following b and c, it can be seen that many controls are missing to be able to affirm that there are differences between the two groups and to what extent this can be attributed to the COVID-19 situation, a circumstance that in any case both groups of police experienced at the academy, or at least that is my understanding.

4.                The discussion makes claims that are difficult to extract from the data obtained and especially from the research design used to obtain them. I do not think it can be claimed that they study changes in the mental health of police trainees according to changes or modifications in the educational environment of the academy and COVID-19. They themselves recognise this aspect in their limitations. Therefore, the discussion needs to be reframed again and grounded in their results. In addition, they should try to compare their results with those obtained in other research more akin to the profession that is the subject of the study sample.

5.                Following all that has been said up to this point, I consider that they have many more limitations than those suggested by the authors.

6.                The conclusions section should be reformulated according to the modifications made in the article.

Round 2

Reviewer 2 Report

Comments and Suggestions for Authors

I would like to thank you for the possibility of being able to revise the new version of this manuscript again, because I have been able, by reading the authors' letter, to get rid of some doubts that arose in the first revision. I consider, upon reading the new revision of the manuscript, that the authors respond and include part of the suggestions indicated by the reviewers. The new elements incorporated by the authors clarify and help to better understand the manuscript and correct some of the methodological deficiencies they presented. Therefore, I believe that the manuscript meets the minimum requirements to be published in the journal.